# Prioritizing sequence variants in conserved non-coding elements in the chicken genome using chCADD

**Christian Groß**[1,2☯]*, **Chiara Bortoluzzi**[3☯], **Dick de Ridder**[1], **Hendrik-Jan Megens**[3], **Martien A. M. Groenen**[3], **Marcel Reinders**[2], **Mirte Bosse**[3]*

**1** Bioinformatics Group, Wageningen University & Research, 6708 PB, Wageningen, The Netherlands, **2** Delft Bioinformatics Lab, University of Technology Delft, 2600 GA, Delft, The Netherlands, **3** Animal Breeding and Genomics Group, Wageningen University & Research, 6708 PB, Wageningen, The Netherlands

☯ These authors contributed equally to this work.
* gross.christian2@gmail.com (CG); mirte.bosse@wur.nl (MB)

**Data Availability Statement:** Raw sequences of the 169 individuals used in this study are available at the European Nucleotide Archive under accession number PRJEB34245 and PRJEB36674.

## Abstract

The availability of genomes for many species has advanced our understanding of the non-protein-coding fraction of the genome. Comparative genomics has proven itself to be an invaluable approach for the systematic, genome-wide identification of conserved non-protein-coding elements (CNEs). However, for many non-mammalian model species, including chicken, our capability to interpret the functional importance of variants overlapping CNEs has been limited by current genomic annotations, which rely on a single information type (e.g. conservation). We here studied CNEs in chicken using a combination of population genomics and comparative genomics. To investigate the functional importance of variants found in CNEs we develop a ch(icken) Combined Annotation-Dependent Depletion (chCADD) model, a variant effect prediction tool first introduced for humans and later on for mouse and pig. We show that 73 Mb of the chicken genome has been conserved across more than 280 million years of vertebrate evolution. The vast majority of the conserved elements are in non-protein-coding regions, which display SNP densities and allele frequency distributions characteristic of genomic regions constrained by purifying selection. By annotating SNPs with the chCADD score we are able to pinpoint specific subregions of the CNEs to be of higher functional importance, as supported by SNPs found in these subregions are associated with known disease genes in humans, mice, and rats. Taken together, our findings indicate that CNEs harbor variants of functional significance that should be object of further investigation along with protein-coding mutations. We therefore anticipate chCADD to be of great use to the scientific community and breeding companies in future functional studies in chicken.

## Author summary

Chickens are raised worldwide as a livestock species to provide us with their eggs and meat, but besides their huge economical impact their genome remains poorly understood.

chCADD scores together with a script to annotate SNPs can be downloaded from the Open Science Framework project page (https://osf.io/8gdk9/). The fully annotated CNEs and a description of the column labels can be downloaded from the same page together with an ad-hoc script to reproduce the SNP density bar plots shown in Fig 5 in the main manuscript. Instructions on how to unpack the files are provided in the ReadMe.txt.

**Funding:** C.G. was funded by the TTW-Breed4Food Partnership, project number 14283: From sequence to phenotype: detecting deleterious variation by prediction of functionality. C.B was funded by the European Union´s Horizon 2020 Research and Innovation Programme under the Grant Agreement No. 677353 (Innovative Management of Animal Genetic Resources – IMAGE). M.B. is financially supported by the NWO-VENI grant no.016.Veni.181.050. The funders had no role in study design, data collection and analysis, decision to publish, or preparation of the manuscript.

**Competing interests:** The authors have declared that no competing interests exist.

Here we introduce a variant prioritization tool modeled after the Combined Annotation Dependent Depletion (CADD). CADD is a well-established approach to prioritize variants with respect to their deleteriousness for the interpretation of genetic variation that can substantially impact human phenotypes, such as diseases. We applied the CADD approach to chicken (chCADD) to investigate the functional importance of conserved non-protein-coding elements. The chCADD model assigns a score to all possible variation in the chicken genome, which can be used to prioritize genetic variants to be used in for breedings strategies. We used these scores to identify subregions within conserved non-protein-coding elements of relative higher importance. The chCADD score and the identified subregions are expected to support our efforts to pinpoint causal genomic variation throughout the chicken genome.

## Introduction

The rapidly increasing availability of genomes has considerably advanced our understanding of the non-protein-coding fraction of the genome. With the sequencing of the human genome [1] and the first ENCODE project [2, 3] it was soon realized that protein-coding genes constitute a small fraction of a species functional genome and that the remaining non-protein-coding DNA is not simply ´junk´ DNA as initially thought. Nevertheless, the functional importance of these non-protein-coding regions remained for long time unknown, as determining (molecular) function was far more difficult than for protein-coding genes [4]. A better understanding of the functional importance of these non-protein-coding regions comes from comparative genomics, which has allowed the systematic, genome-wide identification of conserved non-protein-coding elements (CNEs) [5, 6].

Comparative genomics relies on the genome comparison of a group of species related by a narrow or wide time-scale (i.e. phylogenetic scope). Regions in the genome that share some minimum sequence similarity across two or more species are an indication of a selection constraint. Moreover, conservation often implies a biological function [7]. Based on this principle, CNEs can be identified in any species included in the alignment, as reported in recent studies in the collared flycatcher [8], fruit flies [9], and plants [6]. However, the phylogenetic scope [10] and species included in the alignment [11] can have important implications for the identification of CNEs. For instance, by including the spotted gar genome in their alignment, Braasch et al. (2016) were able to identify numerous CNEs previously undetectable in direct human-teleost comparisons, supporting the importance of a bridging species in the alignment [11].

CNEs have been the subject of intense recent interest. The identification of CNEs has had important implications in enhancing genome annotation [12], investigating signatures of adaptive evolution [13–15], and identifying putative trait loci [16]. CNEs and sequence conservation have also proven crucial in studying the genetic basis of phenotypic diversity. In fact, non-protein-coding SNPs have been linked to traits and diseases in genome-wide association studies [17, 18].

Although the methodological advantages of a comparative genomic approach are well recognized, the functional interpretation of CNEs is incomplete if based on conservation alone, as conservation provides information on restrictions, but not on functionality. A possible solution is combining conservation with other complementary types of data that characterize the biological role of genetic sequences at a genome-wide scale [7]. Such data include, for instance, RNA sequencing (RNA-seq) for the identification of transcriptionally active regions and

chromatin immunoprecipitation followed by sequencing (ChIP-seq) for regulatory-factor-binding regions (RFBRs) [19]. In human genetics, integrative annotations such as Combined Annotation-Dependent Depletion (CADD) [20] have been developed. The main advantage of such frameworks is the combination, into a unique score, of diverse genomic features derived from, among others, gene model annotations, evolutionary constraints, epigenetic measurements, and functional predictions [21].

Compared to humans, for many non-mammalian model species, including chicken (*Gallus gallus*), the situation is quite different. First, comparative genomic studies that made use of the very first genome assemblies [22–24] may have provided an incomplete and biased picture of avian CNEs and avian genome evolution, as recently pointed out by Bornelov et al. (2017) [25]. Second, the lack of species-specific methods that can identify and score functional non-protein-coding mutations throughout the genome has restricted most of the research interest to protein-coding genes. In fact, in the context of protein-coding genes generic predictors such as SIFT [26], PolyPhen2 [27], and Provean [28] can be used.

We here addressed these limitations using a combination of comparative genomic and population genomic approaches to accurately predict CNEs in the chicken genome. Furthermore, we used machine learning to develop a ch(icken) Combined Annotation-Dependent Depletion (chCADD) model, in the tradition of previous CADD models for non-human species, including mouse (mCADD) [29] and pig (pCADD) [30]. As we show, chCADD has the potential of providing new insights into the functional role of non-protein-coding regions of the chicken genome at a single base pair resolution.

Even though deciphering the function of the non-protein-coding portion of a species genome has been a challenging task, we expect our study to provide a new framework for decoding the still largely unknown function of CNEs and their relative variants in chicken, an ideal non-mammalian model and anchor species in evolutionary studies.

## Materials and methods

### Chicken genomic data

We used a dataset by Bortoluzzi and colleagues available at the European Nucleotide Archive (http://www.ebi.ac.uk/ena/) under accession number PRJEB34245 [31] and PRJEB36674 [18]. The dataset comprised a total of 169 individuals sampled from 88 traditional chicken breeds of divergent demographic and selection history. The 169 chicken samples were sequenced at the French Institute of Agricultural Research (INRAe), France, on an Illumina HiSeq 3000. Reads were processed following standard bioinformatics pipelines. Reads were aligned to the chicken GRCg6a reference genome (GenBank Accession: GCA_000002315.5) with the Burrows-Wheeler alignment (BWA-mem) algorithm v0.7.17 [32]. After removal of duplicate reads with the *markdup* option in sambamba v0.6.3 [33], we performed population-based variant calling in Freebayes [34] using the following settings: (1) mapping quality > 20, (2) base quality > 20, (3) at least 20% of observations and 2 reads supporting an alternative allele within an individual, and (4) coverage at SNP position > 4 and < 2.5*average individual genome-wide coverage. We reduced the false discovery rate by additional filtering using BCFtools v1.4.1 [32]. The settings were: (1) a phred quality score > 30, (2) an allele count supporting the alternative allele > 2, (3) maximum number of 10 alleles, (4) variants located within 3 bp of an indel.

### Multiple whole-genome sequence alignment

Conserved elements (CE) were identified using the 23 sauropsids multiple whole-genome sequence alignment (MSA) generated using Progressive Cactus (https://github.com/glennhickey/progressiveCactus) [35] by Green et al. (2014) [36]. The MSA downloaded in the

hierarchical alignment format (HAL) was converted into multiple alignment format (MAF) using the HAL tools command hal2maf [37] with the following parameters: -refGenome galGal4 (GenBank Accession: GCA_000002315.2) to extract alignments referenced to the chicken genome assembly, -noAncestors to exclude any ancestral sequence reconstruction, -onlyOrthologs to include only sequences orthologous to chicken, and -noDupes to ignore paralogy edges. During reformatting, only blocks of sequences where chicken aligned to at least two other species were considered for a total chicken genome alignability of 90.88%. Genomic coordinates were converted to the GRCg6a genome assembly using the pyliftover library in python v3.6.3.

## Prediction of evolutionarily conserved elements

Conserved elements were predicted from the whole-genome alignment using PhastCons [38]. We chose PhastCons because this approach does not use a fixed-size window approach, but can take advantage of the fact that most functional regions involve several consecutive sites [39]. We first generated a neutral evolutionary model from the 114,709 four-fold degenerate (4D) sites previously extracted from the alignment by Green et al. (2014). The topology of the phylogeny was also identical to that derived by Green et al. (2014). PhastCons was run using the set of parameters used by the UCSC genome browser to produce the 'most conserved' tracks (top 5% of the conserved genome): expected length = 45, target coverage = 0.3, and rho = 0.31 [40]. Conserved elements were subsequently excluded if falling in or overlapping assembly gaps and/or if their size was < 4 bp.

## Annotation of conserved elements by genomic feature

We use the Ensembl (release 95) chicken genome annotation files to extract sequence coordinates of CDS, exons, 5' and 3' UTRs, pseudogenes, and lncRNAs. Sequence information was extracted from 14,828 genes (out of the 15,636 genes found in the Ensembl annotation), as transcripts of these genes had a properly annotated start and stop codon. For protein-coding genes with an annotated 5' UTR of at least 15 bp, the promoter was defined as the 2-kb region upstream of the transcription start site (TSS) [8]. Sequence coordinates of miR-NAs, rRNAs, snoRNAs, snRNAs, ncRNAs, tRNAs, and scRNAs were also extracted from the annotation file. For the identification of intergenic regions we considered all annotated protein-coding genes and defined intergenic regions as DNA regions located between genes that did not overlap any protein-coding genes in either of the DNA strands. The intersection between CEs and the various annotated genomic features was found following the approach of Lindblad-Toh et al. (2011) of assigning a CE overlapping two or more genomic features to a single one in a hierarchical format: CDS, 5' UTR, 3' UTR, promoter, RNA genes, lncRNA, intronic, and intergenic region [12]. Conserved non-protein-coding elements (CNEs) were defined as CEs without any overlap with exon-associated features (CDS, 5' UTR, 3' UTR, promoter, and RNA genes) and include lncRNAs, introns, and intergenic regions.

## Gene ontology analysis

Genes in conserved regions overlapping CDS, 5' UTR, 3' UTR, and introns were separately used to perform a Gene Ontology analysis in g:Profiler [41] using *Gallus gallus* as organism. We only considered annotated genes that passed Bonferroni correction for multiple testing with a threshold < 0.05.

## Genome-wide distribution and density of conserved non-protein-coding regions

Polymorphic, bi-allelic SNPs belonging to all functional classes predicted by the Variant Effect Predictor (VEP) [42] were considered. However, to improve the reliability of the set of annotated variants, we applied additional filtering steps. SNPs were discarded if they overlapped repetitive elements or if their call rate was < 70%. The rationale for excluding variants found in repetitive elements was to reduce erroneous functional predictions as a result of mapping issues, as regions enriched for repetitive elements are usually difficult to assemble. For intronic and intergenic SNPs, SNPs in exons or that fell within any spliced EST from the UCSC chrN_intronEST tables were discarded [43].

## Ancestral allele and derived allele frequency

The sequence of the inferred ancestor between chicken and turkey (*Meleagris gallopavo*; Turkey_2.01) [44] reconstructed from the Ensembl EPO 4 sauropsids alignment (release 95) was used to determine the ancestral and derived state of an allele, along with its derived allele frequency. We considered only SNPs for which either the reference or alternative allele matched the ancestral allele. Ancestral alleles that did not match either chicken allele were discarded. We generated derived allele frequency (DAF) distributions for sets of SNPs based on functional class and whether they were within or outside of CNEs. A derived allele frequency cutoff of 10% was used to distinguish rare from common SNPs.

## Chicken Combined Annotation Dependent Depletion (chCADD)

The chicken CADD score is the -10 log relative rank of all possible alternative alleles of all autosomes and Z chromosome of the chicken GRCg6a reference genome, according to the following formula:

$$chCADD_i = -10log_{10}\frac{n_i}{N} \tag{1}$$

where $N$ represents the number of all possible alternative alleles (3,073,805,640) on the investigated chromosomes and $n$ is the rank of the $i$th SNP. The rank is based on the model posteriors of a ridge penalized logistic regression model trained to classify simulated and derived SNPs.

Chicken derived SNPs were defined as those sites where the chicken reference genome differs from the chicken-turkey ancestral genome inferred from the Ensembl EPO 4 taxa alignment containing chicken, turkey, zebra finch (*Taeniopygia guttata*; taeGut3.2.4) [45] and green anole lizard (*Anolis carolinensis*; AnoCar2.0) [46]. Sites for which the ancestral allele occurs at a minor allele frequency greater than 5% were excluded. In addition, derived SNPs that are observed with frequency above 90% in our population of 169 individuals were included. In total we identified 17,237,778 SNPs.

The dataset of simulated variants was simulated based on derived nucleotide substitution rates between the different inferred ancestors of the 4 species in the EPO 4 taxa sauropsids alignment. These derived nucleotide substitution rates were obtained for windows of 100 kb and used to simulate de novo variants which have a larger probability to have a deleterious effect than the set of derived variants. All SNPs which have a known ancestral site are retained in the dataset. In total 17,233,727 SNPs were simulated in this way. 17,233,722 SNPs of each dataset were joined and randomly assigned to train and test sets of sizes 15,667,020 and 1,566,702, respectively.

The datasets were annotated with various genomic annotations: among others, PhyloP and PhastCons conservation scores based on three differently deep phylogenies (i.e. 4 sauropsids,

37 amniote/mammalia, 77 vertebrate, all excluding the chicken genome), secondary DNA structure predictions [47], Ensembl Consequence predictions, amino acid substitution scores such as Grantham [48], and amino acid substitution deleterious scores such as SIFT [49]. Further, we utilized RNA expression, ATAC-seq and Hi-C [50] data to annotate our data set. An overview is given in S1 Table.

Annotations for which values were missing were imputed (S1 Table) and categorical values were one hot-encoded [51]. In the one hot-encoding process, an annotation is a series of binary annotations, each indicating the presence of a specific category for a given variant. For scores that are by definition not available for certain parts of the genome, such as SIFT which is found only for missense mutations, columns indicating their availability were introduced.

Combinations of annotations were created of Ensembl Variant Effect Predictor consequences and other annotations, such as distance to transcription start site and conservation scores. The total number of all features used in training was 874. An extensive list of all annotations, combinations of annotations and their learned model weights is shown in S1 File. Finally, each feature column is scaled by its standard deviation. The logistic regression is trained via the Python Graphlab module. We selected a penalization term of 1 based on results on the test set (S1 Fig).

### Investigation of likely causal SNPs from the OMIA database

We downloaded the likely causal variants of phenotype changes from the Online Mendelian Inheritance in Animals (OMIA) [52] database (last accessed 25.11.2019). SNPs whose location was reported for older genome assemblies such as Galgal4 and Galgal5 were mapped to the chicken GRCg6a reference genome via CrossMap [53]. We only considered bi-allelic SNPs whose genomic position was successfully mapped to GRCg6a and whose substitution remained the same. In total, 15 SNPs were left and annotated with chCADD.

### Change point analysis

To identify subregions of particular importance within each CE, we annotated all CEs with the maximum chCADD score found at each site or the 23 sauropsids PhastCons scores that were used to identify conserved elements in the first place. Our basic assumption was that highly important subregions within a CE are preceded and succeeded by less important sites which would result in a relatively higher score region surrounded by two lower scored regions. Each CE was treated similarly to time series data by conducting an offline change point analysis, once based on maximum chCADD scores and once based on 23 sauropsids PhastCons scores. To this end, we used the Python ruptures module [54] and applied a binary segmentation algorithm with radial basis function (RBF). The algorithm first identifies a single change point. Furthermore, if a change point is detected, the algorithm investigates each sub-sequence independently to identify the next change point We were looking particularly for 2 change points, which would divide the CE into three subregions, numbered from 1 to 3, starting at the 5' end of the sequence. We added 5 bp upstream and downstream of each CE to allow that the borders of the 2nd region coincide with the borders of the CE (Fig 1). After computing the change points, we conducted t-tests between the scores of the 1st and 2nd, as well as 3rd and 2nd subregions, to identify CEs that have a significantly different score in the 2nd section than in the other two. We applied a p-value cutoff of 0.05. We sorted CNEs with respect to the largest difference between the mean chCADD score of the inner and the two outer subregions and selected those with a higher scored 2nd section than either of the other two outer ones.

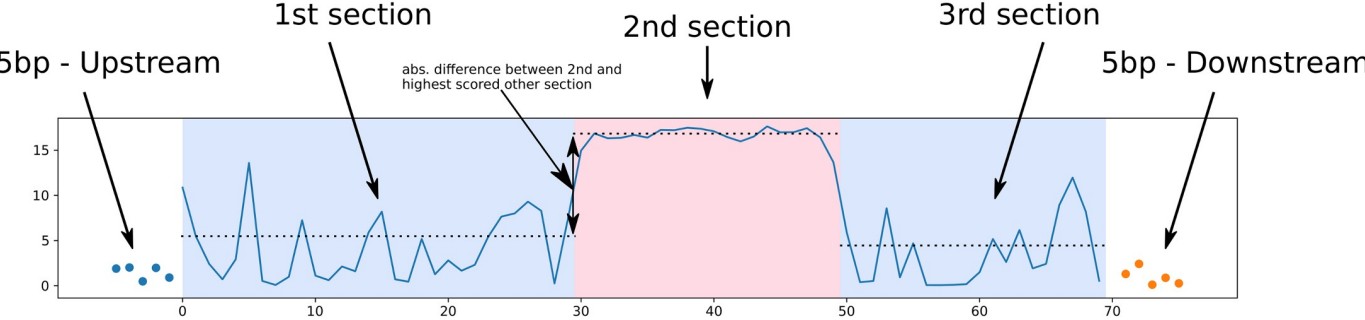

**Fig 1. Approach used to identify subregions within CNEs via change point analysis.** The scores used to annotate the CE region are displayed on the y-axis. The position in the investigated CE region is shown on the x-axis. In total there are five sections, 5 bp up and downstream, 1st, 2nd and 3rd subregions. The transitions from blue to red background indicate the position of the two identified change points. The up and downstream scores are shown as dots while the scores in the CE region are a continuous blue line.

### SNP density distribution within conserved non-protein-coding regions

SNP density was calculated as the number of SNPs identified in the 169 chicken individuals divided by the number of bases found in the sequence. SNP density was computed for conserved coding (CC) and conserved non-protein-coding (CNE) regions, as well as for the subregions identified in the change point analysis of CNEs overlapping lncRNAs, introns, and intergenic regions. We repeated this analysis once for the change points identified using chCADD scores and once for the 23 sauropsids PhastCons based change points.

### Homologous phenotypes

We obtained phenotypes from the Ensembl database (release 95) for genes associated with the lncRNA and intronic CNEs. Beside chicken, these phenotypes encompass the observed phenotypes for orthologous genes associated with disease studies in humans (*Homo sapiens*) and gene-knockout studies in mouse (*Mus musculus*) and rat (*Rattus norvegicus*).

## Results

### Conserved non-protein-coding elements cover a large fraction of the chicken genome

To define CNEs, we first identified conserved elements (CEs) using the UCSC PhastCons most conserved track approach [40]. PhastCons predicted in the 23 sauropsids multiple sequence alignment (MSA) 1.14 million CEs encompassing 8% of the chicken genome for a total of 73 Mb. In line with the density of genes and regulatory features characteristic of the chicken genome [55], we found that most of the predicted CEs are on micro-chromosomes (GGA11-GGA33), followed by intermediate (GGA6-GGA10) and macro-chromosomes (GGA1-GGA5) (S2 Fig). Even though the length of predicted CEs ranged from 4 bp to a maximum of 2,000 bp, the vast majority was short (< 100 bp) (S3 Fig). Therefore, we do not expect any length bias in our final set of CEs.

We annotated CEs by genomic features, considering only genes for which the transcript had a proper annotated start and stop codon, as defined by the Ensembl´s annotation files (n = 14,828 genes). Overall, we found that 23% of the predicted CEs were associated with exonic sequences (i.e. CDS, 5' UTR, 3' UTR, promoter, and RNA genes) spanning 17.14 Mb of the chicken genome (Table 1). The majority of the exon-associated CEs overlapped known coding regions (85% of total exon-associated CEs), followed by 3' UTRs (8% of total), and

**Table 1. Statistics of predicted conserved elements (CEs) based on genomic feature.** The fraction of CEs per sites class is presented, for protein-coding gene annotations, in percentages of the exonic CEs (17,148,879 bp). For non-protein-coding gene annotations, the fraction is relative to the non-exonic CEs (51,224,645 bp). Abbreviations: CC, conserved coding; CNE, conserved non-protein-coding elements.

| Genomic feature | No. overlapping CEs | Total overlap (bp) | Genome coverage (%) | Fraction of site class conserved (%) |
|---|---|---|---|---|
| CDS | 213,787 | 14,683,183 | 1.38 | 85.62 |
| 5'UTRs | 5,457 | 207,320 | 0.02 | 1.21 |
| 3'UTRs | 23,721 | 1,460,144 | 0.15 | 8.51 |
| Promoters | 16,022 | 761,504 | 0.08 | 4.44 |
| RNA genes | 701 | 36,728 | 0.00 | 0.21 |
| lncRNAs | 121,840 | 7,696,557 | 0.80 | 15.03 |
| Introns | 328,579 | 18,520,675 | 1.93 | 36.16 |
| Intergenic | 400,501 | 25,007,413 | 2.60 | 48.82 |
| **Total CC** | **259,688** | **17,148,879** | **1.78** | **100.00** |
| **Total CNE** | **850,920** | **51,224,645** | **5.33** | **100.00** |

promoter regions (4% of total). Although we observed conservation in exon sequences, most CEs overlapped non-protein-coding sequences, including lncRNA (15% of total non-exon associated CEs), intronic (36% of total), and intergenic regions (49% of total). We further examined the biological processes and molecular functions of known genes overlapped by CEs in coding regions, 5' UTRs, 3' UTRs, and introns. These genes are associated with basic functions, including cell differentiation and development, anatomical structure development, morphogenesis, and growth (S2 Table). Most of these GO categories have also been previously associated with mammalian and vertebrate ultraconserved elements (UCEs) [55, 56].

In total we identified 259,688 CEs in protein-coding regions, leaving 850,920 CNEs spanning over 51 Mb of the chicken genome (Table 1), with a genome-wide distribution of 92.10 CNEs/100-kb. We further observed noticeable differences in the length distribution of CEs associated with different types of annotations. Among the conserved exon-associated CEs, those found in CDSs are, on average, the longest (68 bp), followed by 3' UTRs (61 bp), RNA genes (52 bp), promoters (47 bp), and 5' UTRs (38 bp) (S4 Fig). On the contrary, CEs found in non-protein-coding regions show a homogenous length distribution, ranging from 56 bp in introns to 63 bp in lncRNAs (S5 Fig).

## CNEs are less common in gene dense regions

We further investigated the genomic location of CNEs as this might provide important clues to their functional role. We found that the distribution of CNEs in windows of 100 kb is significantly negatively correlated (r = -0.22; *p-value*: < 2.2x10-16) with the distribution of exons (Fig 2a). The correlation between CNEs and exons remained negative even after scaling the CNE count within each window to the remaining sequence length after substracting the coding sequences (Fig 2b). We subsequently analyzed chicken polymorphism data to address the mutational or evolutionary forces shaping CNEs, following previous studies in humans [43] and *Drosophila* [9, 57]. We used polymorphism densities to investigate whether these forces could still be acting on the chicken genome or they could have acted in other species and may no longer be relevant for chicken. SNP density, which reflects events within the chicken lineage, was calculated in the genomes of 169 chickens from different traditional breeds of divergent demographic and selection history. Specifically, we compared the SNP density found in CNEs with that in non-protein-coding elements that were identified not to be conserved (non-CNEs; i.e. not conserved intronic, lncRNA and intergenic regions), following [43, 57]. Overall, we found that the SNP density in non-CNEs (= 0.02) is two-fold higher than CNEs (= 0.01).

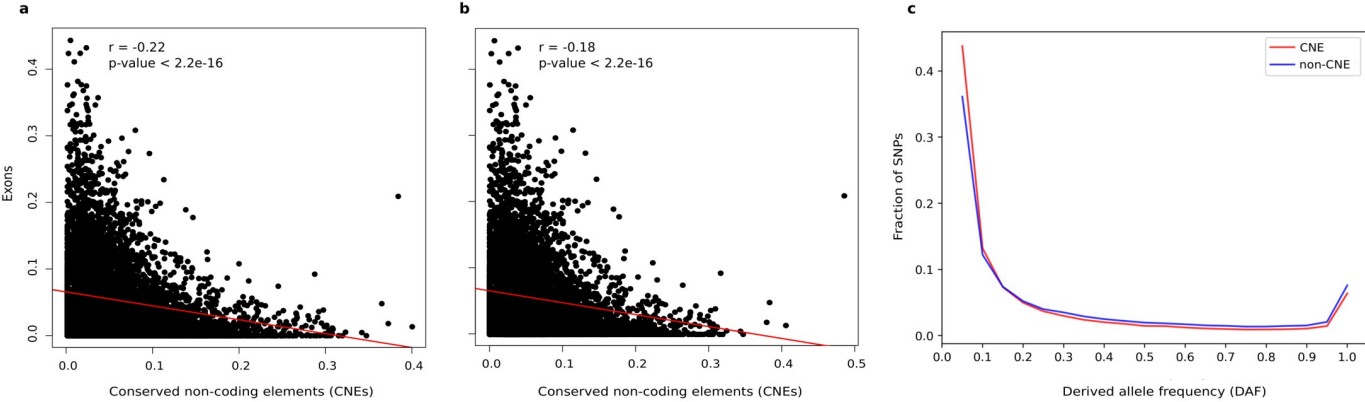

**Fig 2. Conserved non-protein-coding elements are less common in gene dense regions and are selectively constrained.** Correlation between exons and CNEs along the chicken genome when considering (**a**.) and excluding (**b**.) coding sequences. CNEs and exon density was calculated in windows of 100 kb. The Pearson correlation coefficient r and corresponding p-value are shown in the top left corner. **c**. Derived allele frequency (DAF) distribution of SNPs in CNEs and non-CNEs.

## CNEs are selectively constrained in chicken

To test whether low local mutation rates in CNEs or purifying selection is responsible for the observed low SNP density, we looked at the derived allele frequency (DAF) distribution in CNEs and non-CNEs. This is because mutation rate differences are not expected to affect the allele frequency spectra. On the contrary, selective constraint is responsible for the shift in allele frequency distribution of constrained alleles towards lower values. Allele frequencies for derived (new) alleles were compiled using the sequence of the inferred ancestor between chicken and turkey. The ancestral allele was determined for a total of 9 million SNPs that passed several filtering criteria (see Methods). We observed an excess of rare ($\leq$ 10%) derived alleles of SNPs within CNEs in all chicken populations (Fig 2c). Overall, 57% of SNPs within CNEs had a DAF $\leq$ 10%, compared to only 48% in non-CNEs (the same pattern was observed for each SNP functional class; see also Table 2). Non-CNEs displayed on the contrary a higher proportion of common SNPs (DAF > 10%) (52% versus 43% within CNEs) independently of their functional class (Fig 2c; Table 2). Therefore, the lower proportion of derived alleles in CNEs indicates that evolutionary pressure has suppressed CNE-derived allele frequencies.

**Table 2. Derived allele frequency distribution for SNPs in CNEs and non-CNEs.** The derived allele frequency was compiled using the sequence of the inferred ancestor between chicken and turkey. A derived allele frequency of 10% is used as a cut-off to define rare versus common variants. Information are reported for each genomic feature that make up CNEs and non-CNEs.

| Genomic feature | DAF | Within CNEs | Outside CNEs | chCADD within CNEs | chCADD outside CNEs |
|---|---|---|---|---|---|
| | | Number of SNPs (%) | Number of SNPs (%) | Average (± sd) | Average (± sd) |
| All | $\leq$0.10 | 137,871 (57%) | 482,685 (48.4%) | 9.78 (4.18) | 3.21 (3.18) |
| | > 0.10 | 103,726 (43%) | 513,935 (51.5%) | 8.81 (4.25) | 2.74 (2.83) |
| lncRNA | $\leq$0.10 | 24,364 (57.4%) | 26,429 (47.6%) | 10.02 (4.00) | 3.49 (3.33) |
| | > 0.10 | 18,081 (42.5%) | 29,014 (52.4%) | 9.10 (4.13) | 3.03 (2.99) |
| Intron | $\leq$0.10 | 43,790 (56.8%) | 159,203 (47.4%) | 9.81 (4.46) | 3.00 (3.11) |
| | > 0.10 | 33,171 (43.2%) | 176,650 (52.6%) | 8.71 (4.53) | 2.46 (2.74) |
| Intergenic | $\leq$0.10 | 69,717 (57%) | 297,053 (44.6%) | 9.68 (4.05) | 3.31 (3.20) |
| | > 0.10 | 52,474 (43%) | 308,271 (55,4%) | 8.78 (4.11) | 2.87 (2.86) |

## chCADD scores for the investigation of CNE and SNP evaluation

To investigate CNEs further, we developed a model that can evaluate individual SNPs or entire sequences based on a per-base score, with respect to its putative deleteriousness. This model is based on the CADD approach, hence it is labeled ch(icken) CADD. chCADD is a linear logistic model that is trained to differentiate between two classes of variants, one being relatively more enriched in potentially deleterious variants than the other. To obtain these two classes, one class is generated from derived variants, alleles that have accumulated since the last ancestor with turkey and became fixed or almost fixed (allele frequency > 90%) in our chicken populations. These are depleted in deleterious variants and can be assumed to be benign or at least neutral in their nature. The set of putative deleterious variants contains simulated *de novo* variants that are not depleted of deleterious variants. The feature weights obtained during training are shown in S1 File. Performance on a held out test set to determine an optimal penalization term are shown in S1 Fig.

## chCADD distinguishes between potentially causal and non-causal variants

We evaluated the performance and applicability of chCADD on two different sets of variants before we annotated non-coding SNPs. First, we assigned a chCADD score to all SNPs found in the genomes of the 169 chickens previously used in the SNP density and DAF analysis and compared these to functional predictions as annotated by the Ensembl VEP (Fig 3). To this end, we categorized VEP predictions into 14 categories (S3 Table) and joined them to two sets, indicating if they are located in coding or non coding region. The purpose of this was to test whether chCADD correctly scores SNPs with respect to their potential to cause a deleterious or phenotype-changing effect, as indicated (mostly for protein-coding mutations) by the VEP

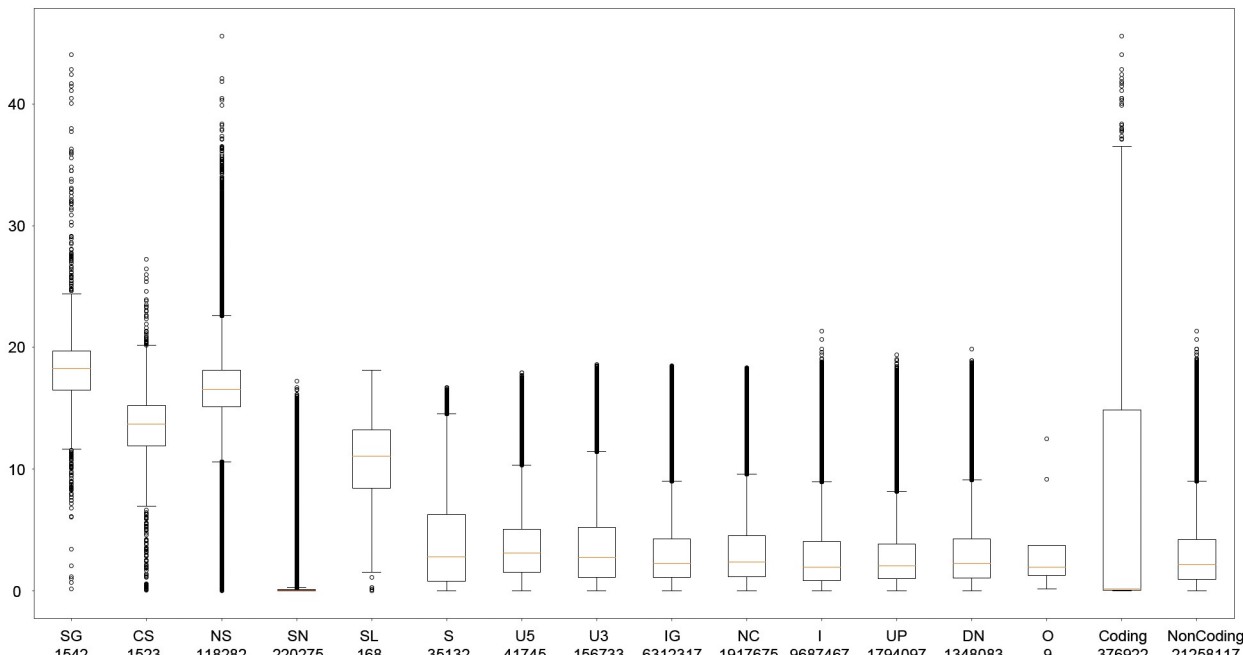

**Fig 3. chCADD score distribution of SNPs per Variant Effect Predictor (VEP) cateogy.** SNPs from the 169 chickens are categorized based on the VEP categories reported in S2 Table. In addition, we joined them based on if they are located within a coding or non-coding region. Abbreviations: SG: Stop-Gained; CS: Canonical Splice; NS: Non-Synonymous; SN: Synonymous; SL: Stop-Lost; S: Splice Site; U5: 5'-UTR; U3: 3'-UTR; IG: Intergenic; NC: Noncoding-Change; I: Intronic; UP: Upstream; DN: Downstream; O: Other. Coding: the joined set of CS, NS, S, SG, SL, SN; NonCoding: DN, I, IG, NC, U3, U5, UP. The label indicates the category and the number of SNPs falling into that category.

functional predictions. We observed that mutations with a relatively large deleterious potential, such as stop-gained mutations and splice-site altering mutations, were scored higher than regular missense and synonymous mutations (Fig 3). SNPs in potentially regulatory active regions were also evaluated to be potentially more deleterious than synonymous SNPs (Fig 3). We performed a similar analysis considering only protein-coding and regulatory mutations found in the Online Mendelian Inheritance in Animals (OMIA) database (Table 3). We annotated only SNPs whose genomic positions were uniquely mapped to the chicken GRCg6a reference genome and the reference/alternative allele matched that in the genome assembly. Of the 15 annotated SNPs associated with a change of phenotype, 5 were reported to cause a deleterious phenotype change in thex affected individual, and an average chCADD score of 27.1. These 5 variants (3 stop-gained, 2 missense) have a chCADD score above 20 and are putatively responsible for dwarfism, scaleless, analphalipoproteinaemia, muscular dystrophy, and wingless phenotypes (Table 3). All these phenotypes display a strong severity and may lead to an early death in uncontrolled environments.

## chCADD detects evolutionary constraints within CNEs

As we showed, chCADD can score functionally important protein-coding variants. We therefore decided to take a step further by annotating SNPs found in CNEs with chCADD to predict their deleteriousness and function (Table 2). We assume that highly scored SNPs can help us to identify truly functionally active regions among CNEs. We observed that rare non-protein-coding variants located within CNEs (DAF $\leq$ 10%) have an overall higher chCADD score compared to rare variants found in non-CNEs (Table 2). This result supports our previous conclusion based on the derived allele frequency spectrum that evolutionarily conserved non-protein-coding variants are likely functional. As expected, this trend was most pronounced in lncRNAs, followed by introns and intergenic regions.

We further used the chCADD score to identify specific subregions of potentially higher functional importance within each CNE, assuming that the high scoring SNPs would indicate that. We applied a change point analysis to search for a center region that has high chCADD scores as opposed to the two outer regions (see Methods). We ranked CNEs based on positive

**Table 3. Annotation of known causative variants with the chCADD score.** SNPs were obtained from the Online Mendelian Inheritance in Animals (OMIA) and their genomic position was lifted over to the GRCg6a reference genome.

| OMIA ID(s) | Variant Phenotype | Gene | Type of Variant | Deleterious? | g. or m. | chCADD |
|---|---|---|---|---|---|---|
| OMIA 0016229031 | Resistance to avian sarcoma and leukosis viruses, subgroup C | BTN1A1 | stop-gain | no | 28:g.903289G>T | 17.8 |
| OMIA 0008899031 | Scaleless | FGF20 | stop-gain | yes | 4:g.63270401A>T | 33.0 |
| OMIA 0015349031 | Resistance to myxovirus | MX1 | missense | no | 1:g.110260061G>A | 14.2 |
| OMIA 0009159031 | Feather colour, silver | SLC45A2 | missense | no | Z:g.10336596G>T | 21.2 |
| OMIA 0009159031 | Feather colour, silver | SLC45A2 | missense | no | Z:g.10340909T>C | 15.7 |
| OMIA 0006799031 | Muscular dystrophy | WWP1 | missense | yes | 2:g.123014353G>A | 26.3 |
| OMIA 0003039031 | Dwarfism, autosomal | C1H12ORF23 | stop-gain | yes | 1:g.53638233C>T | 35.3 |
| OMIA 0013029031 | Resistance to avian sarcoma and leukosis viruses, subgroup B | TNFRSF10B | stop-gain | no | 22:g.1418711C>T | 17.6 |
| OMIA 0008109031 | Polydactyly | LMBR1 | regulatory | yes | 2:g.8553470G>T | 17.4 |
| OMIA 0009139031 | Silky/Silkie feathering | PDSS2 | regulatory | unknown | 3:g.67850419C>G | 3.9 |
| OMIA 0015479031 | Wingless-2 | RAF1 | stop-gain | yes | 12:g.5374854G>A | 23.4 |
| OMIA 0003749031 | Feather colour, extended black | MC1R | missense | no | 11:g.18840857T>C | 18.0 |
| OMIA 0003749031 | Feather colour, extended black | MC1R | missense | no | 11:g.18840919G>A | 18.9 |
| OMIA 0003749031 | Feather colour, buttercup | MC1R | missense | no | 11:g.18841289A>C | 17.4 |
| OMIA 0003749031 | Feather colour, extended black | MC1R | regulatory | no | 11:g.18840609C>T | 6.7 |

chCADD score differences between the center region and the outer regions and filtered for significant difference ($p\text{-}value \leq 0.05$, t-test). The top 3 ranked CNEs that overlap with lncRNAs, intronic and intergenic regions, respectively, are shown in Fig 4a.1, 4b.1 and 4c.1.

Analogous to this subregion analysis based on chCADD score, we performed a subregion analysis based on the 23 sauropsids PhastCons scores. Fig 4a.2–4c.2 show the identified regions for the PhastCons score for the same CNEs as Fig 4a.1 and 4c.1, respectively. These figures indicate that chCADD generates more discriminative subregions than PhastCons. Particularly interesting are the chCADD scores for the top intergenic regions Fig 4c.1). The chCADD score increased from 5 to 15 at the subregion change point. This is equal to an increase of predicted deleteriousness by one magnitude, from the top 33% highest scored sites in the entire genome to the top 3%.

To further investigate the subregion partitioning of the CNEs, we computed the SNP density in each region for CNEs for which we can assume that our assumption of three subregions holds. We did this for both the chCADD induced regions (Fig 5, blue bars) as well as the 23 sauropsids PhastCons induced regions (Fig 5, orange bars). All CNE subregions display an intriguiging difference in SNP density between the upstream and downstream 5bp of the CNE, for which, however, we did not find any explanation (e.g. there is no difference in GC, CpG or open chromatin distributions).

## Conserved non-protein-coding subregions are detected on the basis of a limited number of genomic annotations

As part of the investigation into subregions we identified two change points, splitting each CE into three subregions, starting from 5' to 3', 1*st*-, 2*nd*- and 3*rd* subregion (Fig 1). Next we were interested how genomic annotations that were used in the creation of chCADD, differ between the three subregions. The model coefficients with the largest weights (S4 Table) point to the importance of the PhastCons conservation scores calculated on the 4 sauropsids alignment. Other important model features are secondary structure predictions and combinations with the intronic identifier from VEP. Over all CNEs, we compared the chCADD model features, especially the conservation scores that are based on different phylogenies, excluding the chicken reference sequence in their computation. For all genomic annotations, we computed absolute Cohen's D values (standardized mean difference) [58]. We observed that the conservation scores based on the largest 77 vertebrate alignments cannot properly distinguish between the 1*st*-,2*nd*- and 3*rd* subregions. Conservation scores based on smaller phylogenies (4 sauropsids and 37 amniote/mammalia) are more discriminative between these (S5 Table); see columns 1*st*-2*nd*, 2*nd*-3*rd*).

Considering the three PhastCons scores, based on differently large phylogenies, the average absolute Cohen's D between the 1*st*- and 2*nd*- and the 2*nd*- to the 3*rd*- subregions differ less between different genomic features (intergenic, lncRNA and introns) than between genomic annotations (S5 Table; see columns 1*st*-2*nd*, 2*nd*-3*rd*). The average absolute Cohen's D between the three subregions of a CNE ranges from 0.259 to 0.276. In comparison, the average absolute Cohen's D between the same subregions, taking the three conservation scores individually, range from 0.137 to 0.338. The effect sizes between the different multiple sequence alignment PhastCons score (i.e. 4 sauropsids, 37 amniote/mammalia, 77 vertebrates) differ by more than 2-fold.

## Intronic CNEs overlap functionally important genes

Intronic CNEs were associated with genes for which we obtained phenotype annotations of their orthologs in human, mouse, and rat. We investigated the top 10 CNEs that are located in

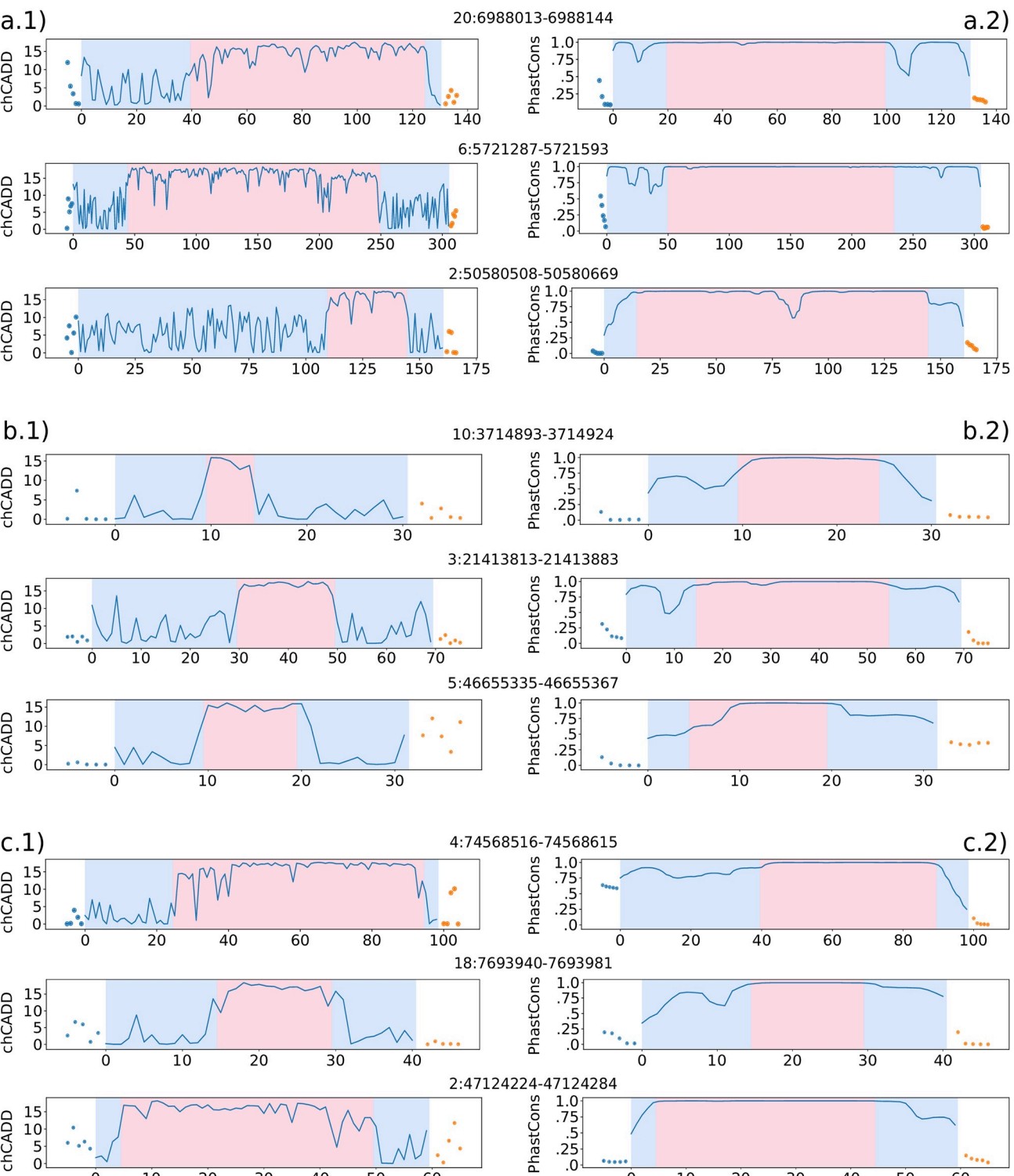

**Fig 4. Change point analysis of the top 3 CNEs for each genomic feature, respectively (lncRNA, intronic, intergenic).** CNEs are sorted based on the largest difference between the 2nd section and 1st or 3rd section for each of the three CNE classes respectively (lncRNA, intronic, intergenic). Change points were once computed based on maximum chADD score per site (a.1,b.1,c.1) and once on 23 sauropsids PhastCons scores (a.2,b.2,c.2). The dots in each plot display the scores for the 5 bp up- and downstream regions. The transition from blue to red background indicates the identified change points. a.1) lncRNA—maximum chCADD a.2) lncRNA—PhastCons scores. b.1) intronic—maximum chCADD. b.2) intronic—PhastCons. c.1) intergenic—maximum chCADD. c.2) intergenic—PhastCons.

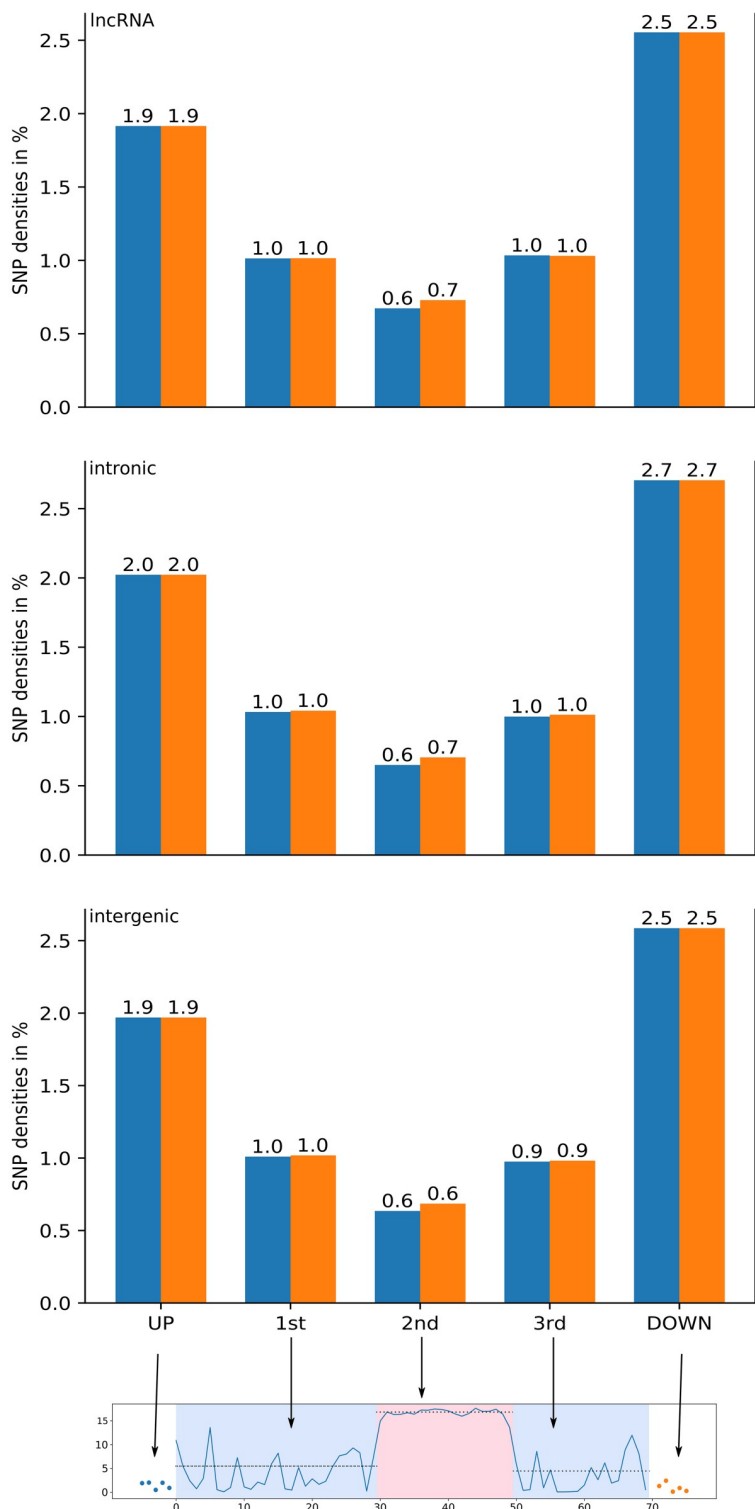

**Fig 5. SNP densities computed for each section of the three different CNEs (lncRNA, intronic, intergenic).** The orange bars represent the SNP densities for that section based on change points derived from the 23 sauropsids alignment PhastCons scores, the blue bars represent the SNP densities based on change points identified via chCADD.

introns, with the largest p-value differences between the 1*st* and 3*rd* to the 2*nd* section. In total, 6 CNEs were associated with homologous genes that have annotated phenotypes in other species. Among the phenotypes found for human genes are mental retardation and non-syndromic male infertility. For mouse, these included neuronal issues and abnormal shape of heart and limbs (S2 File). The link to highly severe phenotypes in other species highlights the potential importance of regulatory features for orthologous genes in chicken.

## Discussion

### The prediction of CNEs depend on the phylogenetic scope

Non-protein-coding elements are typically identified by sequence-level similarity across species, which is a generally applicable criterion of conservation and biological function [10]. However, when predicting CEs, and subsequently CNEs, the evolutionary distance among species included in the alignment (or phylogenetic scope) is an important parameter that can considerably affect the prediction and resolution of CEs. If the evolutionary distance among species is too narrow, the specificity of constraint is reduced, but if it is too broad, the number of CEs rapidly declines and lineage-specific conservation is lost [10, 59].

One of the first studies to address the impact of the phylogenetic scope on CEs prediction was that of Lindblad-Toh et al. (2011). In their study on the 29 mammalian multiple sequence alignment, the authors identified 3.6 million conserved elements spanning 4.2% of the genome at a resolution of 12 bp [12]. When comparing these results to a 5 vertebrate alignment, Lindblad-Toh and colleagues observed that only 45% of the 5 taxa CEs were covered by the 29 taxa alignment. The partial overlap indicates that most of the CEs derived from the 29 taxa alignment were mammalian-specific [12]. The issue resulting from a broad phylogenetic scope on CNEs has also recently been reported by Babarinde and Saitou (2016), where authors identified CNEs between chicken and four mammalian species, including human, mouse, dog, and cattle [60]. By applying a minimum length of 100 bp, Babarinde and Saitou (2016) identified 21,584 CNEs in chicken, a small number as expected from the divergence time between human and chicken occurred approximately 310 million years ago [55]. Therefore, CNEs detected among distant species are better predictions of ultraconserved CNEs than CNEs between closely related species (i.e. human-mouse) [61], as they were already present in the ancient common ancestor of the considered species.

In this study we chose the 23 sauropsids multiple sequence alignment for two reasons. First, the phylogenetic distance between crocodilian and bird species (240 million years ago) [36] is large enough to detect likely functional CNEs. Second, the alignment is reference free allowing the identification of lineage-specific CEs. Reference-free alignments should always be preferred over reference-based ones [62]. In fact, genomic regions shared within a certain clade, which would be missed in a reference-based alignment (e.g. MULTIZ), can also be detected. As a result, reference free alignments better enable the study of genome evolution along all phylogenetic branches equally.

### Avian genomes have similar genomic characteristics

According to our study, 8% of the chicken genome is covered by CEs for a total of 1.14 million CEs. These results are comparable to those on the collared flycatcher genome (*Ficedula albicollis*) [8]. By means of the same alignment, Craig et al. (2018) identified 1.28 million CEs covering 7% of the flycatcher genome. The genome of many bird species is highly compact and thus small in size. Small genomes are thought to require fewer regulatory sequences involved in the organization of chromatin structure [8]. However, the similarity in genome size between, for

example, chicken (i.e. GRCg6a: 1.13 Gb) and flycatcher (i.e. FicAlb1.5: 1.11 Gb), reflects the little cross-species variation characteristic of birds [63].

The limited number of CEs often identified in birds relative to mammals has repeatedly been linked to gene loss [22, 24, 64]. However, the role of gene loss in avian evolution, genome size, and prediction of CEs has recently been questioned. According to Bornelov et al. (2017), gene loss was incorrectly hypothesized from the absence of genes clustering in GC-rich regions in the earlier chicken genome assemblies [25]. In fact, these regions are often difficult to sequence and assemble. The issue is particularly prominent in the GC-rich micro-chromosomes, which, as we show, contribute disproportionately to the total density of functional sequence (S2 Fig). We therefore recommend future comparative genomics studies in chicken to make use of the most recent and complete genome assembly to avoid any erroneous link of CEs to gene loss in chicken.

## Conserved non-protein-coding elements are maintained by purifying selection

A fundamental question in the study of CNEs is the role of purifying selection. Purifying selection can be discriminated from a low mutation rate by comparing the derived allele frequency (DAF) spectra in constrained regions (i.e. CNEs) with that of neutral regions (i.e. non-CNEs) [9, 43]. The rationale is that new mutations are unlikely to increase in frequency in constrained regions. Although CNEs are identified using an interspecific comparative genomic approach, the evolution and dynamics of these regions are generally analyzed at an intraspecific scale by looking at polymorphism data [43, 65]. In this study, we showed that the evolutionary constraint acting on the 23 sauropsids is correlated with constraint within the chicken populations, as assessed from chicken polymorphism data. Consistent with studies in humans [12, 43], plants [6], and *Drosophila* [9, 57], the derived allele frequency spectra of our chicken populations is shifted towards an excess of rare variants in CNEs. These results indicate that the conservation of CNEs in the chicken genome is mainly driven by selective constraints, and not by local variation in mutation rate. The role of purifying selection was also confirmed by the reduced SNP density in CNEs compared to non-CNEs and by the reduced SNP density in specific conserved non-protein-coding subregions. The concordance in SNP density is a clear indication of reduced levels of population diversity and functional roles of CNEs as confirmed by the association of subregions within CNEs to highly severe phenotypes in humans, mouse, and rat. However, future population diversity comparisons in terms of nucleotide diversity ($\pi$) [66] or Watterson's estimator ($\theta w$) [67] between outbred and inbred populations would further elucidate our understanding of purifying selection in CNEs.

## Integrating comparative and functional genomics into a single score

We developed a ch(icken) Combined Annotation-Dependent Depletion (chCADD) approach that provides scores for all SNPs throughout the chicken genome. These scores are indicative of putative SNP deleteriousness and can be used to prioritize variants. The annotation of chCADD relies on the combination of a diverse set of genomic features, including evolutionary constraints and functional data [20, 21]. Multiple sequence alignments of distantly related species are better suited to differentiate conserved sites that can reliably be used to identify functionally important regions. However, these regions are often large enough to question the functional role of the entire region. Our findings show that chCADD outperforms any conservation-based method alone (e.g. PhastCons) in the identification of functionally important subregions within CNEs. Therefore, methods, such as chCADD, are required to fine-tune in

one step CNEs to identify subregions directly linked to—in some cases deleterious—phenotypes.

According to the authors of the original human CADD [20], SNPs with a score above 20 (i.e. the SNP is among the top 1% highest scored potential SNPs in the genome) could be considered deleterious. This means that the higher the score, the higher the chance the variant has a functional effect or may even be deleterious. When annotating protein-coding and regulatory mutations found in OMIA, we observed that SNPs with a chCADD score of 15 can already be considered functional. Therefore, our findings indicate that by setting an arbitrary threshold of 20 may underestimate the fraction of the genome that is actually functional. This is particularly pronounced when the variants in question are located outside protein-coding regions. Therefore we recommend future chCADD users to evaluate the variants identified in their populations to see if they are particularly highly scored compared to other variants in the same genomic region. Further, the signal to order SNPs of interest is obtained over evolutionary timescale, which means that mutations that would have been deleterious for chicken in the past may not be deleterious for chicken in a commercial environment and vice versa. chCADD supports the ordering of SNPs with respect to their potential interest but for final economical evaluations, further information about each investigated SNP may be required.

## Future uses of chCADD

The high scoring of non-protein-coding variants in subregions of CNEs has important implications for future functional and genome-wide association studies (GWAS) in chicken. A very large fraction of trait- or disease-associated loci identified in GWAS are intronic or intergenic. This is expected considering the preponderance of non-protein-coding SNPs on genotyping arrays [5] or along the genome. However, because of a lack of understanding of the function of non-protein-coding mutations, most of the causal mutations reported in the OMIA database are coding. Moreover, in the presence of non-protein-coding mutations, many studies stop at the general locus or—understandably—assume that the closest neighboring gene is affected. However, these assumptions on genomic distance are simplistic. Our findings in chicken demonstrate that chCADD can accurately pinpoint non-protein and protein-coding variants associated with important phenotypes in chicken. Therefore, we expect future genome-wide association studies combined with chCADD to identify novel causal mutations or substantially narrow down the list of potential causal variants in large quantitative trait loci (QTLs). We also expect chCADD to accelerate the discovery and understanding of the biology and genetic basis of phenotypes.

## Conclusion

Deciphering the function of the non-coding portion of a species genome has been a challenging task. However, the availability of genomes from a great variety of species, along with the development of new computational approaches at the interface of machine learning and bioinformatics, has made this task possible in model and non-model organisms. Our findings indicate that an accurate assessment of selective pressure at individual sites becomes an achievable goal. We have also shown that chCADD is a reliable score for the analysis of non-protein-coding SNPs, which should be targeted along with protein-coding mutations in future genome-wide association studies. We therefore anticipate chCADD to be of great use to the scientific community and breeding companies in future functional studies in chicken.

## Supporting information

**S1 Fig. Model performances measured in Receiver Operator Area under the Curve (ROC-AUC) and log-loss for three different ridge penalization terms (0.1, 1.0, 10.0).** The scale is adjusted to make the differences between the models visible. Penalization of 1 was selected due to the lowest log-loss and largest ROC-AUC.
(PDF)

**S2 Fig. Distribution of conserved elements (CEs) along the chicken genome.** The barplot displays the fraction of the genome per chromosome covered by conserved elements.
(PDF)

**S3 Fig. Frequency size distribution of predicted conserved elements.** The y-axis shows the frequency, while the x-axis the size in base pairs (bp) of the predicted conserved elements.
(PDF)

**S4 Fig. Frequency size distribution of predicted conserved elements overlapping exonic-associated gene annotations.** The exonic-associated conserved elements include CDS, 5'UTR, 3'UTR, and promoter regions.
(PDF)

**S5 Fig. Frequency size distribution of predicted conserved elements overlapping non-protein-coding gene annotations.** The non-protein-coding gene annotations include introns, lncRNA, and intergenic regions.
(PDF)

**S1 Table. List of annotations which form the set of descriptive features for which model weights are learned.** Missing values are imputed via the specified values. Annotations of the type (factor) are OneHotEncoded and combinations between annotations form the final feature set.
(PDF)

**S2 Table. GO term enrichment analysis of exonic-associated CE and intronic CEs.**
(PDF)

**S3 Table. VEP consequences summarized in 14 categories.** If multiple annotations exist for the same variant, the consequence is selected according to the displayed hierarchy, starting at 1 and ending at 14.
(PDF)

**S4 Table. Top 10 model features with the largest assigned weight and their explanations.**
(PDF)

**S5 Table. Differences between genomic annotations utilized for the chCADD model.** Differences are measured in absolute Cohen's D between the different subregions in which each CNEs was subdivided in the change point analysis.
(PDF)

**S1 File. Model coefficients of the trained logistic regression model which is used to score the SNPs with respect to their putative deleteriousness.**
(XLSX)

**S2 File. Phenotypes of homologous genes of the top 10 intronic CNEs.** The top 10 intronic CNEs were selected based on the largest differences between the 1*st* and 3*rd* to the 2*nd* section

within a CNE.
(XLSX)

## Acknowledgments

We would like to thank Ole Madsen and Martijn Derks for the useful discussion. The authors wish to thank the three anonymous reviewers for their thoughful comments and efforts towards improving our manuscript.

## Author Contributions

**Conceptualization:** Christian Groß, Chiara Bortoluzzi, Dick de Ridder, Hendrik-Jan Megens, Martien A. M. Groenen, Marcel Reinders, Mirte Bosse.

**Formal analysis:** Christian Groß, Chiara Bortoluzzi.

**Funding acquisition:** Dick de Ridder, Hendrik-Jan Megens, Martien A. M. Groenen, Marcel Reinders.

**Investigation:** Christian Groß, Chiara Bortoluzzi.

**Methodology:** Christian Groß, Chiara Bortoluzzi, Dick de Ridder, Hendrik-Jan Megens, Mirte Bosse.

**Visualization:** Christian Groß, Chiara Bortoluzzi.

**Writing – original draft:** Christian Groß, Chiara Bortoluzzi.

**Writing – review & editing:** Christian Groß, Chiara Bortoluzzi, Dick de Ridder, Hendrik-Jan Megens, Martien A. M. Groenen, Marcel Reinders, Mirte Bosse.

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
