## [Decision Letter · Decision Letter 0]

3 Jul 2020

Dear Dr Bortoluzzi,

Thank you very much for submitting your Research Article entitled 'Evolutionarily conserved non-protein-coding regions in the chicken genome harbor functionally important variation' to PLOS Genetics. Your manuscript was fully evaluated at the editorial level and by independent peer reviewers. The reviewers appreciated the attention to an important topic but identified some aspects of the manuscript that should be improved.

We therefore ask you to modify the manuscript according to the review recommendations before we can consider your manuscript for acceptance. Your revisions should address the specific points made by each reviewer.

[LINK]

Yours sincerely,

Gregory M. Cooper, PhD

Associate Editor

PLOS Genetics

Gregory Barsh

Editor-in-Chief

PLOS Genetics

Reviewer's Responses to Questions

**Comments to the Authors:**

Reviewer #1: This is a valuable contribution to chicken genomics as the authors have established a database with CADD scores in an attempt to predict the functional significance of sequence variants in the chicken genome. The analysis appears solid and the data are in general well presented. I have the following comments:

1. The title can be improved, I recommend to delete “harbour functionally important variation” because this is the obvious and most common explanation for evolutionary conservation. This study does not provide any functional assessment of non-coding variants so the title becomes misleading. It is highly appropriate to include chCADD in the title.

2. Abstract middle and in several other places: I am missing something after (chCADD) for instance “scores” or “database”.

3. As far as I understand the Author Summary should be popular science-style which is not the case here. For instance, “a change point analysis” is not a common term for laymen.

4. Page 3. Please add a brief explanation how many and which type of breeds were included in the analysis.

Line 272. The title is misleading. It would be more appropriate to say something like “CNEs are less common in gene dense regions”

Line 285-286, the same number of decimal places should be used here and it would be easier for the readers if you write that the SNP density is two-fold higher in non-CNEs (=0.02) than in CNEs (=0.01).

General: It is not clear what you mean by SNP density and you should use average nucleotide diversity instead. This take into account SNP density and allele frequency spectra and should be more powerful for this analysis.

Line 301. It is more appropriate to write “the lower proportion” as the difference is small.

Line 310. “chCADD scores potentially causal variants higher”. Higher compared with what?? My suggesting is to show the distribution of CADD scores for coding and non-coding regions separately in a main figure. Then the authors can judge how extreme the scores presented in Table 3 are. Furthermore, the subtitle is not optimal since the verb scores could be misinterpreted as a substantive.

Table 3. It is sufficient to use one decimal place for the CADD scores.

Figure 1. I think that Figures 1-3 can be merged into one figures with three panels.

Figure 3. Is the relatively high frequency of derived SNPs that are fixed an artefact or is it just a consequence of drift and derived alleles reaching a “point of no return”?

Figure 5. This figure should be based on nucleotide diversity rather than SNP density. A consistent number of decimal places should be used in the figure. The much higher SNP density 5bp downstream than 5bp upstream must be an artefact that the authors need to correct or explain properly.

Reviewer #2: This study was to develop chCADD to identify conserved non-protein-coding regions (CNE) with functionally important variants in chicken genome. The predicted CNEs with potential functional constraint based on chCADD could be useful in narrowing down candidate SNPs for causative mutations in chickens. However, current chCADD model is still suffering from not integrating other important epigenomic data available in chickens (although it is limited compared to human and mouse) as epignomic information could provide more weighted information on the CNE. This is particularly true from the recent 2019 CADD paper (CADD: predicting the deleteriousness of variants throughout the human genome. Philipp Rentzsch et al. Nucleic Acids Research, Volume 47, Issue D1, 08 January 2019, Pages D886–D894).

In addition, several recent studies indicated a considerable CNEs, especially enhancers, might be missed based on sequence conservation across evolutionarily distant species, as they are not necessarily conserved in sequence, but potentially functionally conserved on the regulatory role of the regions across multiple species.

From Table 3, it seems if the phenotype is not deleterious, chCADD score for regulatory variants are very low. does this mean the prediction would not be accurate as SNPs in the regulatory regions in general are not detrimental, but could be functionally important for economically important traits.

The mspt could be improved by comparing annotated regulatory regions from recent published FAANG data in chickens with CNEs identified in this study.

Minor comments: current Ensembl version is 99, not sure why release 95 was used (Line 230). From Line 82, not sure why using galGal4 as more updated version galGal6 is available.

Reviewer #3: In their manuscript "Evolutionarily conserved non-protein-coding regions in the chicken genome harbor functionally important variation" (PGENETICS-D-20-00791) Christian Groß and others describe an analysis of non-protein coding regions in the chicken genome. They use conserved elements identified from multi-species whole genome alignments and they explore how conservation scores or combined annotation scores can be used to break down larger conserved elements into potential functional units. As part of these efforts, the authors release a "Combined Annotation Dependent Depletion" (CADD) model for chicken with this publication. For those unfamiliar, CADD was originally developed for the human genome, where a rich set of available annotation can be combined in a model trained on human derived fixed changes (that were exposed to many generations of purifying selection) vs. simulated "de novo" variants (that have not seen purifying selection). Several of the authors were involved in two previous publications that have extended the development of CADD scores to mouse and pig. While mouse again profits from being a model species, the pig model – similar to the proposed chicken model – needs to rely heavily on gene model derived annotations and species conservation. Many epigenetic features and genomic segmentations are simply unavailable. This is a "the best that we can do" situation and from earlier results as well as the ones presented here, the combined score performs better than just using a single (conservation) metric. Preparing and maintaining scores for three species is also a substantial effort and provides a relevant resource for the respective communities. All three score sets currently provided are limited to SNVs (already in training). Therefore, authors provide scores for all SNVs rather than tools to calculated mCADD, pCADD or chCADD scores.

I only have a number of small comments that I would like the authors to address:

Line 75: What are these filter criteria? Is that what is described in "Genome-wide distribution and density of conserved non-protein-coding regions" it needs a cross reference.

Line 134: Not sure, I understand the intronic EST filter. Please explain what you are addressing here.

Line 135: How can the protein-coding variants fall outside of coding sequence?

Line 163-164: Is that a different alignment than one used above? Why?

Line 177: "imputed" - reference Table S1 for values.

Line 187: Feature scaling. Your scaling as described is alright, just wondering whether you also mean-centered and forgot to mention. Mean centering would only help if one wanted to interpret the model offset, so it is not relevant but commonly done. Also, have you considered releasing the determined SDs and scripts for scoring variants? You might also consider releasing scripts for annotating, deriving training data and scoring variants; in case you are not planning to commit on future score releases (e.g. due to updated reference sequences). Releasing the obtained scores (as already done) is critical; releasing more is of course optional.

Line 274-275: I am trying to wrap my head around that. All originate from a distribution of conserved elements. Wondering whether a random distribution from which you subtract the coding, would always result in a negative correlation with coding, in which case this would probably need a different test.

Figures: Huge issues in PDF conversion. Linked PNGs are much better, but have issues with transparency (tested several programs and did not find one that shows it correctly).

Textual edits:

Abstract: "Depletion (chCADD) [annotation|model],"

Author summary: "The chCADD [model] assign[s] a score"

Line 55: "Depletion (chCADD) model, in the tradition"

Line 100: "falling [in] or overlapping"

**Have all data underlying the figures and results presented in the manuscript been provided?**

Reviewer #1: Yes

Reviewer #2: Yes

Reviewer #3: **No: **I don't see the "fine mapped" CE blocks being released. Without those a number of results or figures could not be reproduced.

PLOS authors have the option to publish the peer review history of their article (what does this mean?). If published, this will include your full peer review and any attached files.

Reviewer #1: No

Reviewer #2: No

Reviewer #3: **Yes: **Martin Kircher

---

## [Editor Report · Decision Letter 1]

5 Aug 2020

Dear Dr Bortoluzzi,

We are pleased to inform you that your manuscript entitled "Prioritizing sequence variants in conserved non-coding elements in the chicken genome using chCADD" has been editorially accepted for publication in PLOS Genetics. Congratulations!

Yours sincerely,

Gregory M. Cooper, PhD

Associate Editor

PLOS Genetics

Gregory Barsh

Editor-in-Chief

PLOS Genetics

Comments from the reviewers (if applicable):

**Data Deposition**

http://datadryad.org/submit?journalID=pgenetics&manu=PGENETICS-D-20-00791R1

**Press Queries**

---

## [Editor Report · Acceptance letter]

18 Sep 2020

PGENETICS-D-20-00791R1

Prioritizing sequence variants in conserved non-coding elements in the chicken genome using chCADD

Dear Dr Bortoluzzi,

We are pleased to inform you that your manuscript entitled "Prioritizing sequence variants in conserved non-coding elements in the chicken genome using chCADD" has been formally accepted for publication in PLOS Genetics! Your manuscript is now with our production department and you will be notified of the publication date in due course.

With kind regards,

Matt Lyles

PLOS Genetics

On behalf of:
